# Long-term associative memory in rats: Effects of familiarization period in object-place-context recognition test

**Shota Shimoda**[1], **Takaaki Ozawa**[2], **Yukio Ichitani**[1], **Kazuo Yamada**[1]*

1 Institute of Psychology and Behavioral Neuroscience, University of Tsukuba, Tsukuba, Ibaraki, Japan,
2 Institute of Protein Research, Osaka University, Suita, Osaka, Japan

* kayadama@human.tsukuba.ac.jp

**Data Availability Statement:** All relevant data are within the manuscript and its Supporting Information files.

## Abstract

Spontaneous recognition tests, which utilize rodents' innate tendency to explore novelty, can evaluate not only simple non-associative recognition memory but also more complex associative memory in animals. In the present study, we investigated whether the length of the object familiarization period (sample phase) improved subsequent novelty discrimination in the spontaneous object, place, and object-place-context (OPC) recognition tests in rats. In the OPC recognition test, rats showed a significant novelty preference only when the familiarization period was 30 min but not when it was 5 min or 15 min. In addition, repeated 30-min familiarization periods extended the significant novelty preference to 72 hours. However, the rats exhibited a successful discrimination between the stayed and replaced objects under 15 min and 30 min familiarization period conditions in the place recognition test and between the novel and familiar objects under all conditions of 5, 15 and 30 min in the object recognition test. Our results suggest that the extension of the familiarization period improves performance in the spontaneous recognition paradigms, and a longer familiarization period is necessary for long-term associative recognition memory than for non-associative memory.

## Introduction

Recognition memory is necessary to discriminate novel information from what is already known. Since animals have an innate tendency to respond to or explore novel stimuli, the habituation-dishabituation paradigm has been regarded as a useful behavioral test to assess recognition memory in various animal species including Aplysia [1], rodents [2], monkeys [3], and humans [4]. In particular, researchers have evaluated rodents' recognition memory using several types of spontaneous recognition tests.

Spontaneous recognition tests have been used to evaluate not only simple non-associative recognition memory (e.g., object recognition test [2]) but also more complex associative recognition memory (e.g., place recognition test [5]; object-context recognition test [6]; object-place-context (OPC) recognition test [7]). While non-associative recognition memory is

**Funding:** This study was supported by grants from JSPS KAKENHI (19K21806 for KY, TO, YI, 19K21808 for TO, 19K03385 for TO, 19H01769 for TO, 19H05005 for TO, 21H00311 for TO, 21K18557 for TO), HOKUTO Foundation for the Promotion of Biological Science (No grant number) for TO, Uehara Memorial Foundation (No grant number) for TO, Takeda Science Foundation (No grant number) for TO, The Mitsubishi Foundation (No. 202011006) for TO, Kowa Life Science Foundation (No grant number) for TO, Research Foundation for Opto-Science and Technology (No grant number) for TO, and the Salt Science Research Foundation (No. 2146) for TO.

**Competing interests:** The authors have declared that no competing interests exist.

**Abbreviations:** OPC, object-place-context; DI, discrimination index.

typically composed of single elements of information such as objects, associative recognition memory is necessary to recognize objects using combined information on multiple elements that typically include the context where animals encountered the objects, as well as the information on the objects and locations.

A standard object recognition test consists of a sample phase and a test phase, with a retention interval inserted between the two phases. In the sample phase, a rat is allowed to explore an open-field arena, in which a pair of two identical objects are placed, for a few minutes for familiarization. After the retention interval, the rat is returned to the arena where one of the objects is replaced with a novel object (test phase). In general, performance in the test depends on the length of the retention interval such as a shorter ($<$ 24 hours) and a longer ($\geqq$ 24 hours) retention interval [2,8, see also 9]. A preferential exploration toward the novel object is defined as a successful discrimination, and the rat is considered to exercise a great ability of recognition memory. Moreover, performance in the spontaneous recognition memory test could be affected by the length of the familiarization period (sample phase). Previous studies systematically examined how the extension of familiarization periods improved performance in the object or place recognition tests. For example, in an object recognition test, animals showed novel object preference in a 5 min familiarization with both short (15 min) and long (24 hours) retention interval [10]. Likewise, in a place recognition test with a long (24 hours) retention interval, a 20 min, but not 5 min familiarization was sufficient for rats to discriminate a replaced object from a stayed one [11].

In more complex associative recognition memory tests, such as the OPC recognition test, however, rats would identify an association between objects, places, and contexts. Typically, the OPC recognition test consists of two sample phases, a test phase, and a retention interval inserted between the second sample and test phases. In the first sample phase, rats are familiarized with two different objects in a context. In the second sample phase, the rats are moved to another context in which the same pair of objects are placed in a swapped position. Subsequently, in the test phase, then, the rats are placed in one of the contexts and allowed to explore a pair of one of the objects that the rats encountered in either the first or the second contexts. In the OPC recognition test, rats are expected to explore the replaced object longer than the one that stayed in each context. Several studies demonstrated that short-term recognition memory had been evident in the OPC recognition test which consisted of 2–5 min familiarizations and 2–15 min retention intervals [7,12–19], although, to our knowledge, long-term recognition memory ($\geqq$24 hours) has not been tested yet.

In the present study, we hypothesized that (1) the extension of the familiarization period in the sample phase facilitated associative recognition memory and enabled animals to exhibit long-term associative recognition memory in the OPC recognition test, and (2) a longer familiarization period was necessary for the formation of long-term associative recognition memory than for the non-associative memory. Here, we systematically investigated the relationship between the lengths of familiarization periods (5, 15, or 30 min) and subsequent novelty discrimination performance in the object, place, or OPC recognition tests in rats (Experiment 1). We also examined how long the associative recognition memory in the OPC recognition test could be retained (Experiment 2).

## Materials and methods

### Experiment 1

**Subjects.** Thirty-two male Long-Evans rats (10–11 weeks old; Institute for Animal Reproduction, Ibaraki, Japan) were used. The mean and standard deviation of their body weight were 355.93 ± 31.54 g at the beginning of behavioral experiments. They were housed

individually and kept on a 12 h light/dark cycle (lights on at 8:00 a.m.) and provided *ad libitum* access to food and water throughout the experiments. All experimental tests were conducted during the light phase. All experiments were approved by the University of Tsukuba Committee on Animal Research.

**Apparatus.** Two open-field arenas (900 × 900 × 450 mm) made of black polyvinyl chloride or white acrylic plexiglass were used to test under two different contexts (S1A and S1B Fig). The black context consisted of a gray floor and black walls. On one of the walls, a white–black vertically striped pattern was attached as a spatial cue. The white context consisted of a white floor and walls. A white–black checkered pattern was attached to one of the walls. The illumination at the center of each arena was 60 lx. An overhead camera was used to record the movement of the rat for the analysis. Background white noise (50 dB) was continuously present during all experimental tests to mask any extraneous noise. The stimulus objects were copies of 10 different objects made of glass, metal, or plastic and varied in height between 7 and 15 cm (S1C Fig). All the objects were adequately heavy or fixed on the heavy metal plate such that the rat could not move them.

**Habituation.** Habituation sessions were conducted for 3 consecutive days. On each day, rats received 5 min of handling by an experimenter, and were then placed in each of the black and white contexts without any objects for 30 min (with at least 60 min interval). The order of the exposure to each context was counterbalanced.

Following the habituation, rats were divided into three groups according to the length of the familiarization periods (5min, *n* = 11; 15min, *n* = 11; 30min, *n* = 10). All rats were subjected to three kinds of spontaneous recognition tests: OPC, place, and object recognition tests. Rats assigned to one or another familiarization condition were subjected to the assigned condition in all tasks' sample phases.

**Object-place-context recognition test.** The OPC recognition test consisted of two sample phases and a test phase (Fig 1A). A 24-hour retention interval was inserted between the first and second sample phases and test phases. In the first sample phase (sample 1), rats were allowed to explore the black context where two different objects were diagonally placed at 22.5 cm apart from the adjacent two walls (e.g., object A on the top left and object B on the bottom right). In the second sample phase (sample 2), the rats were placed at the other white context in which the same pair of objects were placed in a swapped position relative to that in the first sample phase (e.g., object B on the left and object A on the right). Each rat was allowed to explore these objects freely for 5, 15, or 30 min in each sample phase. In the 5-min test phase, rats explored a pair of one of the sample objects (e.g., object A-A) in one of the contexts (e.g., black context). At the beginning of each phase, rats were randomly placed at the one of the four corners facing the walls of the arena. If rats had associative recognition memory of objects, places, and contexts, they would show a preferential exploratory behavior towards the object placed in the novel place-context combination (the dashed arrow in Fig 1A). The positions of the novel objects (e.g., top left or bottom right) in the test phase were counterbalanced. After each phase, the floor of the arena was cleaned using a wet cloth containing sodium hypochlorite solution and the objects were wiped with 70% ethanol to eliminate odor.

**Place recognition test.** Three-seven days after the OPC recognition test, the rats were subjected to a place recognition test. All rats were subjected to re-habituation to the black context for 15 min without any objects. Two identical objects (object C) and the black context were used in this test (Fig 1B). Animals were allowed to explore two objects for 5, 15, or 30 min in the sample phase. After a 24-hour retention interval, the animals were returned to the context in which one of the objects were moved to a novel location and allowed to freely explore for 5 min in the test phase. Note that for the place recognition test, one of the objects was presented in the same location as familiar, whereas the other was moved to a different

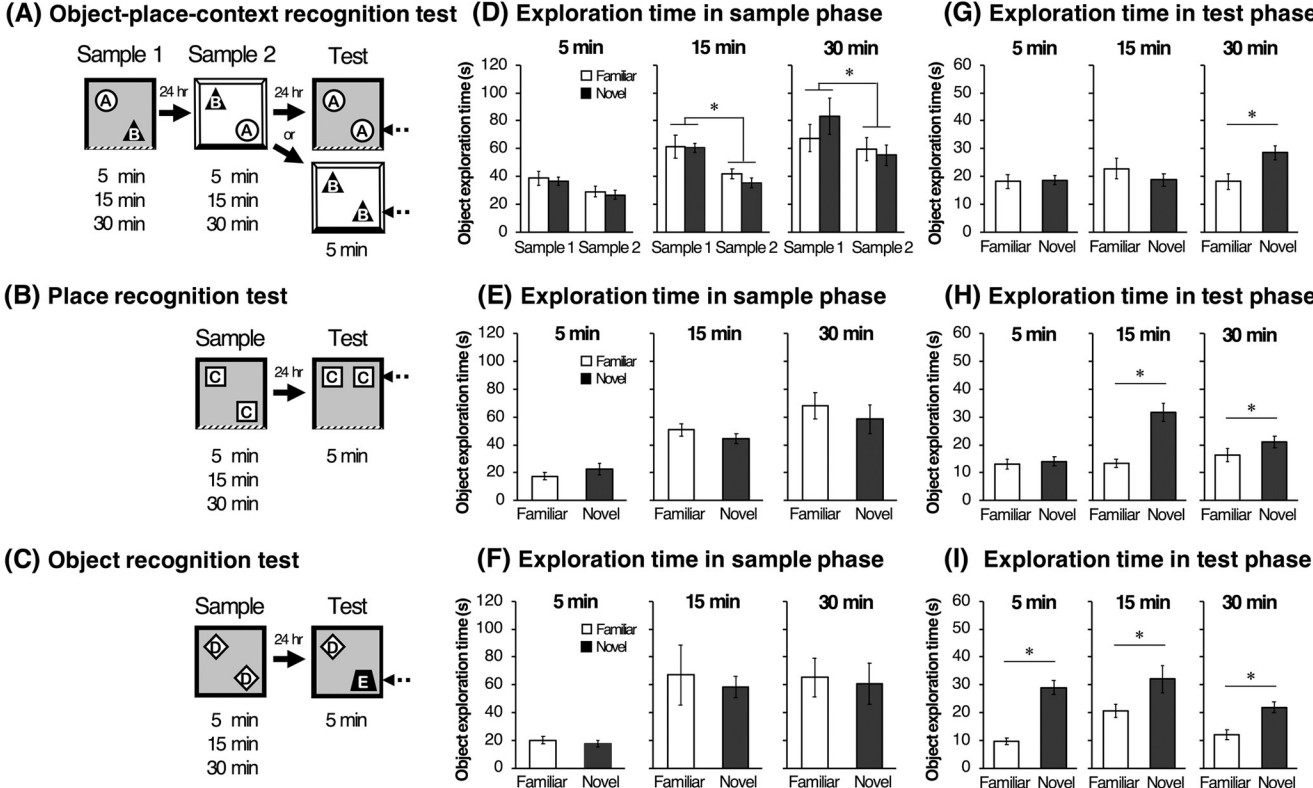

**Fig 1.** Schematic illustrations of object-place-context (OPC) recognition test (A), place recognition test (B), and object recognition test (C). Each test consists of sample phase (5, 15 or 30 min), retention interval (24 hours), and test phase (5 min). A successful discrimination is defined as a preferential exploration of the 'object in a novel context-place combination' in OPC recognition test, 'the object in a novel location' in place recognition test or 'the novel object' in object recognition test and is indicated by dashed arrows. Mean (±SEM) time spent in exploration for each object in sample phase of OPC recognition test (D), place recognition test (E), and object recognition test (F). Mean (±SEM) time spent in exploration for each object in test phase of OPC recognition test (G), place recognition test (H), and object recognition test (I). A two-way (Phase × Object) ANOVA followed by a post-hoc Scheffe test was used for the analysis of the exploration time in sample phases of OPC recognition test. A paired Student's t-test was used for comparison between familiar and novel objects. $^*p < .05$.

location (dashed arrow in Fig 1B), which was placed 30 cm apart from the familiar object and 22.5 cm apart from a sidewall (two locations were possible).

**Object recognition test.** Three-seven days after the place recognition test, rats were tested in an object recognition test (Fig 1C). All rats were subjected to re-habituation in the black context without the spatial cue for 15 min. In the object recognition test, the animals were allowed to explore two identical objects (object D) for 5, 15, or 30min in the sample phase. After the retention interval, animals were returned to the same open-field arena where one of the objects are replaced with a novel object (object E) in the 5 min test phase. The positions of the novel object in the test phase were counterbalanced.

## Experiment 2

**Subjects.** Twenty male Long-Evans rats (7–8 weeks old; Institute for Animal Reproduction, Ibaraki, Japan) were used. The mean and standard deviation of their body weight were 278.59 ± 49.24 g at the beginning of behavioral experiments. Rats were assigned to each experiment (Experiment 2–1, *n* = 10; Experiment 2–2, *n* = 10).

**Object-place-context recognition test with longer retention interval (Experiment 2–1).** The procedure of the OPC recognition test was identical to that used in Experiment 1,

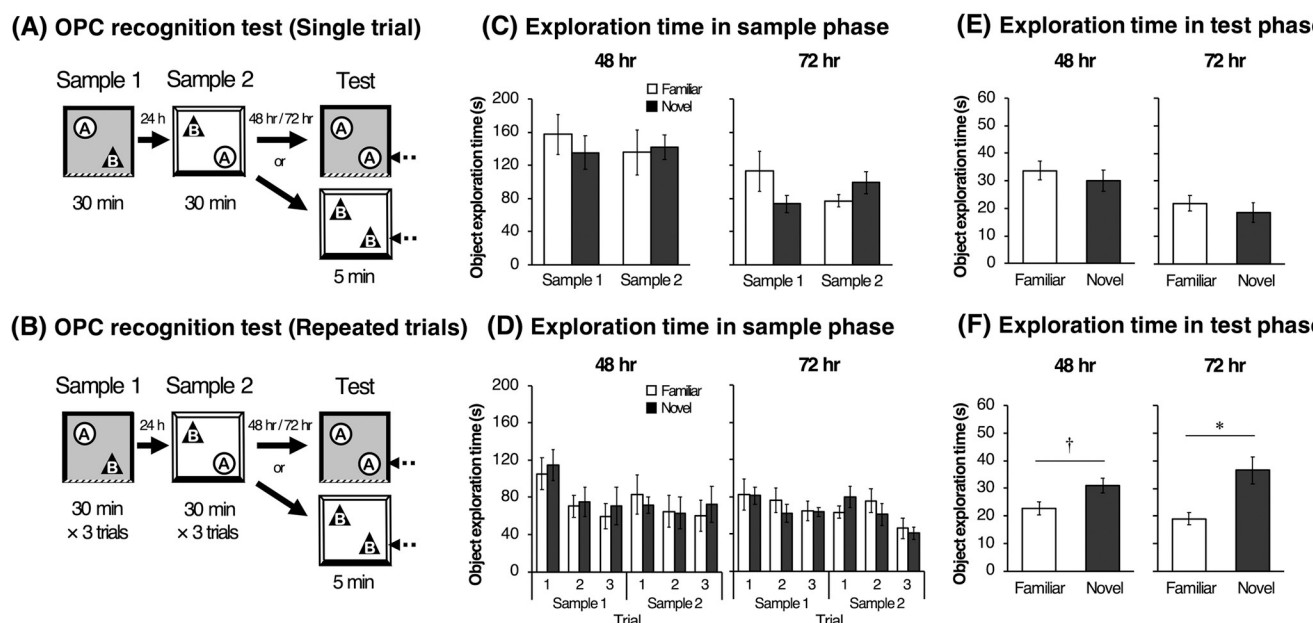

**Fig 2.** Schematic illustrations of OPC recognition test in Experiment 2–1 (A) and 2–2 (B). In each sample phase, rats were subjected to one (Experiment 2–1) or three trials (Experiment 2–2) of the 30-min familiarization. The sample 2 and test phases were separated by either of retention interval conditions (48 or 72 hr). Mean (±SEM) time spent in exploration for each object in sample phases (C, D), in test phase (E, F). A two-way (Phase × Object) and three-way (Phase × Trial × Object) ANOVA followed by a post-hoc Scheffe test was used for the analysis of the exploration time in sample phases. A paired Student's t-test was used for comparison the exploration time between familiar and novel objects in test phase. $^*p < .05$. † $p < .10$.

except that the retention interval was 48 or 72 hours (Fig 2A). All rats were subjected to a 30-min exploration in the sample 1 and 2 phases and a 5-min exploration in test phase following the retention interval (48 or 72 hours). The order of the retention interval conditions was counterbalanced.

**Object-place-context recognition test with repeated familiarization periods (Experiment 2–2).** The procedure of the OPC recognition test was identical to that used in Experiment 2–1, except that rats experienced repeated explorations in the sample phase (Fig 2B). In the sample phase 1 and 2, all rats were subjected to 3 trials of 30-min familiarization with 4-hour inter-trial intervals.

## Data analysis

The ANY-maze video tracking software (Stoelting Co., Illinois, USA) was used to analyze automatically locomotor activity and to count manually the rats' exploratory behavior in each test. In each phase, we manually counted the time rats spent exploring the objects. High inter-rater reliability ($\alpha = .805$) of scoring by two independent judges based on video recorded behaviors of the test phases meant that the assessment procedure was reliable.

Exploration was defined as the rat sniffing, pawing, and directing its nose toward the objects within a distance of 3 cm, except standing over or climbing on the objects. As a measure of discrimination behavior in the test phase, discrimination index (DI) was calculated by dividing the difference in the time spent exploring the novel and familiar by the total time of exploration for both objects [$DI = (T_{novel} - T_{familiar})/(T_{novel} + T_{familiar})$]. A value of zero indicates no preference, while a positive value indicates more exploration of the novel object and a negative value indicates preferential exploration of the familiar object. Exploration time in the

OPC, place recognition, object recognition test at each phase was analyzed using a paired Student's t-test (two-tailed), two-way (Phase × Object) or three-way (Phase × Trial × Object) analysis of variance (ANOVA) followed by a post-hoc Scheffe test. DIs in all conditions at each test were analyzed using a one-way ANOVA. In addition, DIs were also compared to the theoretical chance level (0%) using a one-sample t-test (two-tailed). The correlations between the total exploration time in the sample phase and DI in the test phase were assessed using the Pearson's product-moment correlation coefficients. According to a previous study [15], an exclusion criterion of a statistical outlier was defined as occurring when DI exceeded ± 2 standard deviations from the mean of all rats in each test. If subjects met the criterion, the data in sample and test phases was excluded from the analysis. All values are expressed as mean ± standard error of the mean (SEM). Statistical significance was set at $\alpha$ = 0.05.

## Results

### Experiment 1

**Object-place-context recognition test.** In the OPC recognition test, one subject was excluded from analyses according to the statistical outlier criterion. Fig 1D shows the mean exploration time for the familiar and novel objects in the sample phases 1 and 2. Note that 'familiar' refers to the object that will be the same object and position, and 'novel' refers to the object that will be replaced with a novel object in the test phase. A repeated two-way ANOVA, with the within-subjects variables of Object (familiar vs. novel) and Phase (sample 1 vs. sample 2), showed a significant main effect of Phase in the 30 min [$F(1, 9)$ = 7.91, $p$ = .020] and 15 min [$F(1, 10)$ = 26.97, $p$ = .004] conditions but not in the 5 min condition. A main effect of Object and an interaction between Object and Phase were not significant in each condition. Fig 1G shows the mean exploration time for the familiar and novel objects in the test phase. Paired t-tests revealed that rats explored the novel object significantly more than familiar one ($t(9)$ = 3.14, $p$ = .011) in the 30 min condition. The mean DIs of each condition are shown in Fig 3. A one-way ANOVA showed that there was no significant difference in DIs between each condition. One-sample t-tests revealed that DI was significantly higher than chance level only in the 30 min condition ($t(9)$ = 2.85, $p$ = .018). Pearsons' product-moment correlation coefficients were calculated to determine the relation between performance in the sample and test phases. In the 15 min condition, the total amount of exploration time in the sample phases was positively correlated with DI in the test phase (S2A Fig; $r$ = 0.71, $p$ = .013).

**Place recognition test.** In the place recognition test, one subject was excluded from analyses according to the statistical outlier criterion in the 5 min and 30 min conditions. Fig 1E shows the mean exploration time for the familiar object and novel objects in the sample phase. A paired t-test revealed that there was no significant difference in the exploration time for each object in the sample phase. In the test phase, the mean exploration time for each object (Fig 1H) and DIs in each familiarization condition (Fig 3) were compared. A paired t-test revealed that rats explored the novel object significantly more than the familiar one when the sample phases were 15 min ($t(10)$ = 5.73, $p$ < .001) and 30 min ($t(8)$ = 2.57, $p$ = .032). A one-way ANOVA revealed a significant main effect of Period [$F(2, 27)$ = 14.20, $p$ < .001], and the post hoc test showed that DIs in the 15 min condition were significantly higher than those in the other conditions ($p$ < .001). Furthermore, one-sample t-tests revealed that DIs were significantly higher than chance level in the 15 min ($t(10)$ = 7.66, $p$ < .001) and 30 min ($t(8)$ = 2.83, $p$ = .022) conditions. Significant negative correlation between the total exploration time in the sample phase and DI in the test phase was found in the 30 min condition (S2B Fig; $r$ = -0.71, $p$ = .031).

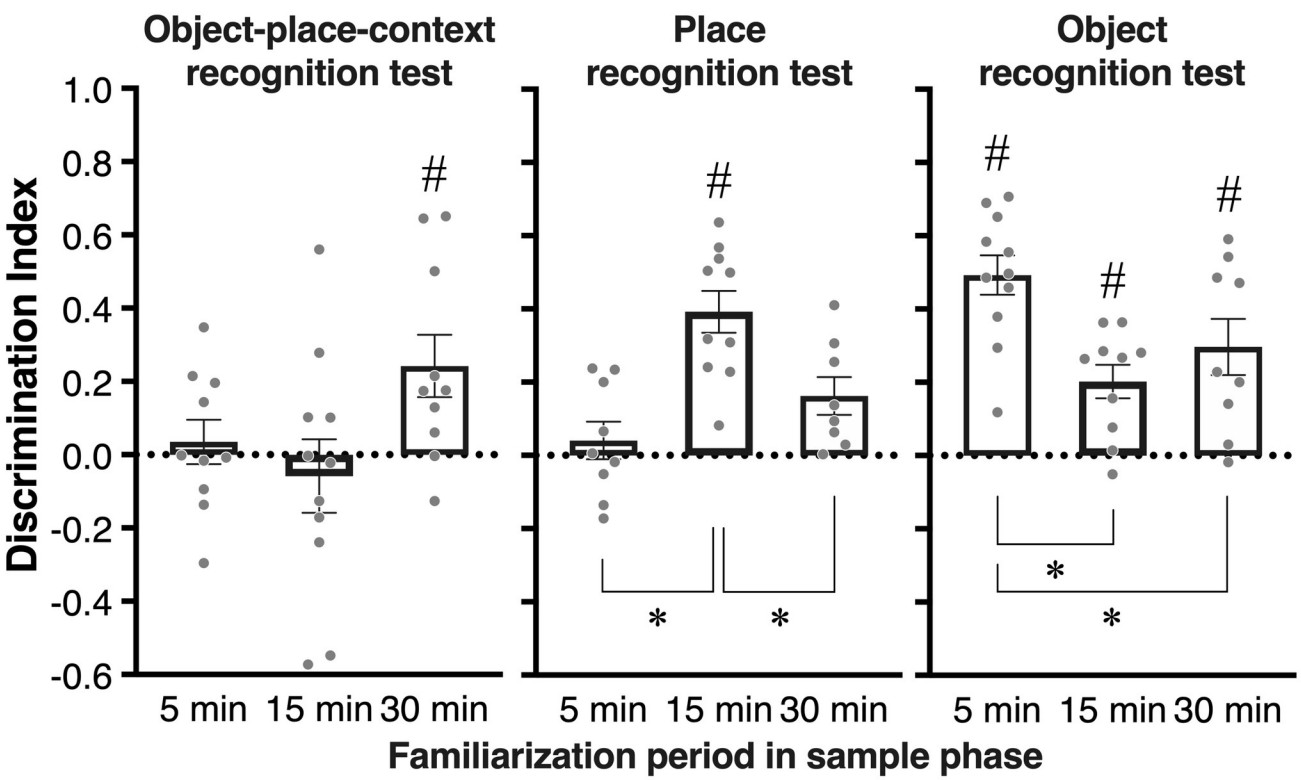

**Fig 3. Mean (±SEM) discrimination index in test phase of OPC recognition test, place recognition test, and object recognition test.** Each individual value is plotted as a gray dot. The horizontal dotted line indicates chance level (0). A one-way ANOVA followed by a post-hoc Scheffe test was used for comparison among each familiarization condition. *$p < .05$. # $p < .05$, † $p < .10$ compared to chance level using one-sample t test.

**Object recognition test.** In the object recognition test, one subject was excluded from analyses according to the statistical outlier criterion in the 15 min and 30 min conditions. Fig 1F shows the mean exploration time for the familiar object and novel objects in the sample phase. A paired t-test revealed that there was no significant difference in the exploration time. In the test phase, the mean exploration time for each object (Fig 1I) and DIs under three different familiarization conditions (Fig 3) were compared. A paired t-test revealed that rats explored the novel object significantly more than the familiar one in all conditions (5 min, $t$ $(10) = 6.74$, $p < .001$; 15 min, $t(9) = 3.32$, $p = .008$; 30 min, $t(8) = 3.48$, $p = .008$). A one-way ANOVA revealed a significant main effect of Period [$F (2, 27) = 6.74$, $p = .004$], and the post hoc test showed that DIs in the 5 min condition were significantly higher than those in the 15 min ($p = .003$) or 30 min condition ($p = .026$). One-sample t-tests also revealed that DIs were significantly higher than chance level in all conditions (5 min, $t(10) = 9.19$, $p < .001$; 15 min, $t$ $(9) = 4.39$, $p = .001$; 30 min, $t(8) = 3.87$, $p = .004$). In the 30 min condition, a significant negative correlation between the total exploration time in the sample phase and DI in the test phase was found (S2B Fig; $r = -0.71$, $p = .029$).

## Experiment 2–1: How long can the associative recognition memory be retained in a single trial of OPC recognition test

In experiment 2–1, one subject was excluded from analyses according to the statistical outlier criterion in the OPC recognition test. Fig 2C shows the mean exploration time for the familiar

## Object-place-context recognition test

**Fig 4. Mean (±SEM) discrimination index in test phase of OPC recognition test in experiment 2–1 and 2–2.** The horizontal dotted line indicates chance level (0). # $p < .05$, † $p < .10$ compared to chance level using one-sample t test.

and novel objects in the sample phases 1 and 2. A repeated two-way ANOVA, with Phase (sample 1 vs. sample 2) × Object (familiar vs. novel) as within-subjects variables, showed no significant main effects or interaction in both the 48 hr and 72 hr conditions. In terms of the mean ± *SEM* exploration times (Fig 2E) and DIs (Fig 4) in test phase, paired t-tests and one-sample t-tests revealed that there was no significant difference. Also, there was no significant correlation between the total exploration time in the sample phase and DI in the test phase of the 48 hr and 72 hr conditions (data not shown).

### Experiment 2–2: Repeated familiarization periods can enhance associative memory in OPC recognition test

Fig 2D shows the mean exploration time toward the familiar and novel objects for three trials in each sample phase. In the 48 hr condition, a repeated three-way ANOVA, with Phase (sample 1 or 2) × Trial (Trial 1, 2 or 3) × Object (familiar or novel) as within-subjects variables, revealed a significant main effect of Phase [$F(1, 9) = 7.95$, $p = .020$] and Trial [$F(2, 18) = 11.55$, $p < .001$] and no interactions. Shaffer's multiple comparison tests revealed that the exploration time in Trial 1 significantly longer than those in Trial 2 and 3 ($ps = .007$). In the 72 hr

condition, a repeated three-way ANOVA only revealed a significant main effect of Trial [$F$(2, 18) = 10.88, $p$ < .001]. Shaffer's multiple comparison tests revealed significant differences between Trial 1 and Trial 2 and between Trial 2 and Trial 3 ($ps$ = .005). Fig 2F shows the mean exploration time for each object in the test phase. Paired t-tests revealed that rats explored the novel object significantly more than the familiar one in the 72 hr condition ($t$(9) = 3.72, $p$ = .004), but not in the 48 hr condition (albeit marginally significant; $t$(9) = 2.00, $p$ = .076). The mean DIs of each condition are shown in Fig 4. One-sample t-tests revealed that DI was significantly higher than chance level in the 72 hr condition ($t$(9) = 4.34, $p$ = .001), but not in the 48 hr condition (albeit marginally significant; $t$(9) = 2.04, $p$ = .070). Furthermore, no significant correlation was found between the total exploration time and DIs at both conditions (data not shown).

## Discussion

In the present study, we investigated the relationship between the length of familiarization at the sample phase (5, 15, 30 min) and subsequent novelty discrimination performance at the test phase in the object, place and OPC recognition tests with a 24-hour retention interval. In the OPC recognition test, rats showed a significant novelty preference when the familiarization period was 30 min, but not when it was 5 min or 15 min (Figs 1G and 3). Furthermore, this novelty preference was retained for 72 hours when the 30-min familiarization period was repeated three times (Figs 2F and 4). In contrast, rats showed successful discrimination even under the shorter familiarization conditions such as 15 min in the place recognition and 5 min in the object recognition (Figs 1H, 1I and 3). These results demonstrated that the long-term (24 hr) associative recognition memory in the OPC recognition test could be evident by extending the familiarization period. Furthermore, it is also suggested that the formation of the long-term complexed associative memory required longer familiarization compared to the non-associative and simple associative memories.

The findings that successful recognition memory was evident in 5-min familiarization in the object recognition test, but not in the place recognition test, are consistent with our previous study showing that rats needed longer familiarization in the place recognition test than the object recognition test [11]. Since rats are required to process the information on both objects and locations in the place recognition, it is reasonable that rats needed more time to process complexed information in the place recognition test than in the object recognition test. Thus, our results showing the relationship between the performance of recognition memory tests and the length of familiarization periods are likely to reflect the differences in difficulty among the object, place and OPC recognition tests. Previous studies reported that primates and humans spent more time gazing at a novel image than a familiar one in the habituation-dishabituation paradigm, and the longer familiarization period is required when the stimulus is more complex [4,20,21]. Although Gaskin et al. [10] demonstrated that a longer exploration of the objects in the familiarization period did not improve non-associative memory performance in the object recognition test of rodents, our findings, which showed that the longer the familiarization period, the more time rats spent in exploration for the objects in the sample phase, demonstrated that the formation of associative recognition memory needs much longer exploration for the objects. These results suggest that a sufficient exploration of the environment, as well as objects and/or locations, can lead animals to make associations between each element of information, such as objects, locations, and contexts.

Interestingly, our results did not show that the longer familiarization period simply improved the recognition performance, suggesting that the appropriate familiarization period may differ depending on the type of recognition memory. This interpretation could be

supported by the results that DIs in the 15 min and 5 min conditions were significantly higher than those in other conditions in the place and object recognition tests, respectively (Fig 3). Also, the negative correlations between the exploration time in the sample phase and the recognition performance were observed in the object and place recognition test (S2B and S2C Fig). As the familiarization period is extended, rats likely focus on the background information of objects, such as locations or contexts. A previous study [10] reported that an excessive amount of object exploration time and repeated trials in the sample phase did not improve the novelty preference in the object recognition test. On the other hand, it is noteworthy that novelty preference in the OPC recognition test is evident by extending the familiarization period, suggesting that a longer familiarization period allows rats to focus on the background information of objects. These results indicate that the appropriate length of the familiarization period may vary from task to task. A bell curve of performance in the object and place recognition tests may reflect that too much familiarization leads to an over-learning, which may cause a decrement of recognition performance.

It should be noted that there were two problems in our experimental design in the OPC recognition test. Firstly, there was inescapably a difference in the retention period between sample 1 (black) and sample 2 (white) contexts. Given that the shorter the retention period is, the stronger the memory strength is, it is likely that the difference in the retention period affects recognition performance. Additionally, in an object-context recognition test, Tam, Bonardi, & Robinson [22] showed that rats exhibited a better recognition performance in the test conducted in the last sample context, suggesting that relative recency could contaminate recognition performance. Secondly, we conducted three kinds of recognition tests in a specific order: OPC, place, and object recognition tests. We cannot exclude the possibility that the prior experience in the OPC recognition test could affect performance in the following object or place recognition tests. This effect is known as a learning set [23], which acquiring a learning strategy in a preceding task improves performance in the following task. For example, Zeldin & Olton [24] demonstrated that a prior spatial learning improved subsequent performance by developing a spatial learning set. Therefore, further studies are needed to elucidate these confounding factors.

Associative recognition memory has been regarded as an episodic-like memory, which is typified as the comprehensive information of "what", "where" and "when" acquired from "a single experience" [7]. Indeed, although several tests have been developed to measure episodic-like memory in rodents, some of them are thought to be inappropriate for episodic-like memory tests. For example, a food reinforcement-based test [25] is unlikely to meet the definition of episodic-like memory because it requires multiple training sessions. According to its definition, the test for episodic-like memory should be completed in a few training sessions. In addition, direct comparison between associative and non-associative memories is thought to be impossible, even in reinforcement-free spontaneous recognition tests, due to differences in the procedures of the tests. The associative memory test (episodic-like memory test [26,27]) used more objects (e.g., 4 objects that have different memory properties including the object, place, and temporal element) in a single-trial test compared to the non-associative memory tests in which two objects are usually used. In other words, the performance in the episodic-like memory test using multiple objects in a single trial could depend on how much animals focus on each memory element (object, place, and temporal) in the test phase. In the OPC recognition test, however, only two objects are used and repeated training is not required in the case of within 24-hour retention interval. Also, it includes the elements of episodic-like memory. Thus, the OPC recognition test can be the more appropriate paradigm for a rodents' episodic-like memory test, and it can systematically investigate the cognitive and neural

mechanisms in both associative and non-associative memory by combining with object recognition and place recognition tests.

In conclusion, our results showed that long-term associative recognition memory was evident when the familiarization period was extended to 30 min in the OPC recognition test. We also indicated that repeated 30-min familiarization caused a longer retention of recognition memory in the OPC recognition test. The findings suggested that longer familiarization periods are necessary for the recognition of the complex associative memory compared to simple associative and non-associative memory. We propose that a spontaneous recognition paradigm is a useful tool for the systematic assessment of long-term associative and non-associative recognition memory in rats.

## Supporting information

**S1 Fig.** Apparatus of black context (A), white context (B), and objects (C) used in the experiments.
(TIF)

**S2 Fig.** Correlations between exploration time in sample phase and discrimination index in test phase for the object-place-context (A), place (B), object (C) recognition tests.
(TIF)

## Author Contributions

**Conceptualization:** Shota Shimoda, Takaaki Ozawa, Yukio Ichitani, Kazuo Yamada.

**Data curation:** Shota Shimoda, Takaaki Ozawa, Kazuo Yamada.

**Formal analysis:** Shota Shimoda.

**Funding acquisition:** Takaaki Ozawa, Yukio Ichitani, Kazuo Yamada.

**Investigation:** Shota Shimoda.

**Methodology:** Shota Shimoda, Takaaki Ozawa, Kazuo Yamada.

**Supervision:** Kazuo Yamada.

**Writing – original draft:** Shota Shimoda, Takaaki Ozawa.

**Writing – review & editing:** Yukio Ichitani, Kazuo Yamada.

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
