## [Decision Letter · Decision Letter 0]

5 Mar 2021

PONE-D-21-00442

Long-term associative memory in rats: effects of familiarization period in object-place-context recognition test

PLOS ONE

Dear Dr. Yamada,

Thank you for submitting your manuscript to PLOS ONE. After careful consideration, we feel that it has merit but does not fully meet PLOS ONE’s publication criteria as it currently stands. Therefore, we invite you to submit a revised version of the manuscript that addresses the points raised during the review process.

Both reviewers identified a series of significant problems with your manuscript. Their comments are included. If you are able to address all of these points please resubmit a revised manuscript respecting the points below.

We look forward to receiving your revised manuscript.

Kind regards,

Robert Sutherland, Ph.D

Academic Editor

PLOS ONE

Journal Requirements:

Reviewers' comments:

Reviewer's Responses to Questions

**Comments to the Author**

1. Is the manuscript technically sound, and do the data support the conclusions?

Reviewer #1: Partly

Reviewer #2: Partly

2. Has the statistical analysis been performed appropriately and rigorously? 

Reviewer #1: Yes

Reviewer #2: Yes

3. Have the authors made all data underlying the findings in their manuscript fully available?

Reviewer #1: Yes

Reviewer #2: Yes

4. Is the manuscript presented in an intelligible fashion and written in standard English?

Reviewer #1: Yes

Reviewer #2: Yes

5. Review Comments to the Author

Reviewer #1: Shimoda et al. examined whether increasing the duration of the learning phase would impact retention performance on three versions novelty discrimination task: object, place, and object-place-context (OPC). Their experimental design as their data analyses are suitable. The findings will most likely be pertinent to a small group of researchers, which makes me question whether this manuscript would be better for a more targeted publication venue. Regardless, I have several comments to improve the manuscript:

1- The authors claim that the object and place version of the task do not involve associative learning. These claims need support/references. Actually, I strongly disagree that the place version does not include associative learning. Specifically, learning the location of an object requires associating several configural cues. Moreover, the space version of the task requires contribution form the hippocampus, which is believed to be necessary for configural associations. I suggest that the authors review the literature on configural associations (Rudy and Sutherland 1995) as well as some of studies that have used the place version for higher order cognitive processing (Mumby et al. 2002; Saucier et al. 2008). I will concede, however that the OPC involves more cognitive demand, but, again, a reference supporting the claim would be beneficial.

2- The authors should correlational analyses between investigation time during the sample phase and performance on the retention test.

3- The authors make an interesting argument that the OPC version could be used as an -episodic-like memory task. Such a task becomes of greater use if the memory can be retained for days, weeks, or even months. At the moment, the authors only present evidence that the 30-min sample phase can create a memory good for 24 hours. I suggest that they include a second experiment to assess duration of the memory and whether more than one learning/sample session would be needed for a long-lasting OPC memory. Adding this experiment would greatly improve the value of the study and potential impact.

4- Line 78: “although long-term recognition memory has never been tested.” Is too bold. The authors should alter the sentence to something along these terms: “although, to our knowledge, long-term recognition memory has (≧24 hours) findings have yet to be reported.”

5- The authors should consider including pictures of their testing environments and objects.

6- In the place and object versions, wouldn't an increase in retention performance be expected with the increasing sample phase durations? This is not the case and should be reconciled with the general hypothesis of the study. Also, relating this to the cited work by Dr. Mumby’s lab would be of benefit.

Mumby D, Gaskin S, Glenn M, Schramek T, Lehmann H. 2002. Hippocampal damage and exploratory preferences in rats: Memory for objects, places, and contexts. Learning & Memory 9: 49-57.

Rudy JW, Sutherland RJ. 1995. Configural association theory and the hippocampal formation: an appraisal and reconfiguration. Hippocampus 5: 375-389.

Saucier DM, Shultz SR, Keller AJ, Cook CM, Binsted G. 2008. Sex differences in object location memory and spatial navigation in Long-Evans rats. Anim Cogn 11: 129-137.

Reviewer #2: PONE-D-21-00442

REVIEW:

This series of experiments utilized the spontaneous objects recognition test used in rodents that can be used to assess recognition and more complex associative learning and memory processes. The specific question of interest for the investigators was whether altering the length of the sample phase would impact subsequent novelty discrimination in three different versions of the task including object, place, and object-in-place recognition tasks.

The results showed that increasing the familiarization period improved performance on the different variants of spontaneous recognition tasks, but this procedural manipulation was particularly important for the object-in-place task.

Overall, the experiments were well motivated and executed and the issue of optimal procedures for obtaining reliable results in learning and memory research is an important issue.

INTRODUCTION:

I thought that the introduction was well written.

METHODS, RESULTS and DISCUSSION:

-what effect does the context pre-exposure have on learning these tasks?

-what effect does training the same animals on the three different versions? For example, if another investigator used the parameters used here but only one of the paradigms. Would the effects be the same?

-What about the task design found below? What impact does it have?

“Rats assigned to one or another familiarization condition were subjected to the assigned condition in all tasks' sample phases.”

-line 130-where were the contexts placed (different rooms)? If in the same room, are they placed in different spots in the room. If not, is the path to and the entry to these rooms different (head direction).

-what is meant by place here? Egocentric space or more spatial/relational?

-line 159-“Three-7” is a bit confusing.

-statistics were appropriate and I liked the inclusion if inter-rater reliability.

-line 298-“leaning” should be changed to “learning”

-line 299-“For example, Zeldin & Olton [23] demonstrated that prior

spatial learning improved subsequent memory performance due to proactive interference.”

Why would previous spatial learning improve performance because of proactive interference? Unpack this for the reader.

-one difference between the tasks is the elements are the same on two versions but a completely novel element is introduced in the object recognition task. What impact does this have?

-although mentioned in the discussion, the way that the task were trained sequentially in the same animals might be a significant confound in regards to the results. The concern is if the groups were only trained on one of these tasks with the experimental variable manipulations (sample time exposure) would you get the same effects?

6. PLOS authors have the option to publish the peer review history of their article (what does this mean?). If published, this will include your full peer review and any attached files.

Reviewer #1: No

Reviewer #2: No

---

## [Author Response · Author response to Decision Letter 0]

25 May 2021

Dear Editors and Reviewers:

Thank you for your letter and for the reviewers’ comments concerning our manuscript entitled “Long-term associative memory in rats: effects of familiarization period in object-place-context recognition test” (Manuscript No.: PONE-D-21-00442). We appreciate the time and effort you and the reviewers dedicated to providing feedback on our manuscript. We are grateful for the insightful comments and valuable improvements to our paper. We have carefully studied the comments and have revised the manuscript accordingly. The reviewer comments are laid out below in italicized font, and specific concerns have been numbered. Our responses are given in a point-by-point manner below, and changes are shown in red in the manuscript.

Response to the reviewers’ comments:

Reviewer #1:

Shimoda et al. examined whether increasing the duration of the learning phase would impact retention performance on three versions novelty discrimination task: object, place, and object-place-context (OPC). Their experimental design as their data analyses are suitable. The findings will most likely be pertinent to a small group of researchers, which makes me question whether this manuscript would be better for a more targeted publication venue. Regardless, I have several comments to improve the manuscript:

Response: Thank you very much for your comments and helpful suggestions.

Point 1) The authors claim that the object and place version of the task do not involve associative learning. These claims need support/references. Actually, I strongly disagree that the place version does not include associative learning. Specifically, learning the location of an object requires associating several configural cues. Moreover, the space version of the task requires contribution form the hippocampus, which is believed to be necessary for configural associations. I suggest that the authors review the literature on configural associations (Rudy and Sutherland 1995) as well as some of studies that have used the place version for higher order cognitive processing (Mumby et al. 2002; Saucier et al. 2008). I will concede, however that the OPC involves more cognitive demand, but, again, a reference supporting the claim would be beneficial.

Response 1) We totally agree with your comments. According to the comments, we have redefined the place recognition memory as an associative memory and modified some sentences in the Introduction and Discussion accordingly. Actually, Langston & Wood (2010) reported that rats with hippocampal lesions were unable to discriminate a novel object from a familiar one in an object-place-context recognition test and a replaced object from a stayed one in a more allocentric version of a place recognition test, in which the starting point was changed at the sample and test phase. They also claimed that the rat with hippocampal lesion had an intact discrimination ability in object-context and place recognition memory tests which involved less cognitive demands. In terms of the object-context and place recognition memories, their results are not consistent with Mumby et al. (2002). Based on these findings, we believe that the object-place-context recognition is more complex than the object-context, place and object recognition.

Langston RF, Wood ER. Associative recognition and the hippocampus: Differential effects of hippocampal lesions on object‐place, object‐context and object‐place‐context memory. Hippocampus. 2010;20: 1139–1153. doi:10.1002/hipo.20714

Point 2) The authors should correlational analyses between investigation time during the sample phase and performance on the retention test.

Response 2) Thank you for your suggestion. According to the suggestion, we conducted correlational analyses between the exploration time during the sample phases and performance on the retention test. We have added the descriptions about the correlational analysis in the Results [Object-place-context recognition test, Page 10, Line 253-256; Place recognition test, Page 11, Line 278-280; Object recognition test, Page 11, Line 294-296], in the Discussion [Page 15, Line 380-382], and a figure (Fig S2).

Point 3) The authors make an interesting argument that the OPC version could be used as an -episodic-like memory task. Such a task becomes of greater use if the memory can be retained for days, weeks, or even months. At the moment, the authors only present evidence that the 30-min sample phase can create a memory good for 24 hours. I suggest that they include a second experiment to assess duration of the memory and whether more than one learning/sample session would be needed for a long-lasting OPC memory. Adding this experiment would greatly improve the value of the study and potential impact.

Response 3) Thank you for your valuable suggestions to improve the quality of our manuscript. According to the suggestion, we conducted additional experiments (Experiment 2). In summary, recognition memory in the OPC recognition test with a single trial of 30-min familiarization in the sample phase could not retain for more than 24 hours, but a repeated familiarization extended a successful retention up to 72 hours. We have added descriptions about Experiment 2 (Method [Page 7-8, Line 173-197]; Results [Page 12-13, Line 298-336]) and figures (Figs 3, 4). 

Point 4) Line 78: “although long-term recognition memory has never been tested.” Is too bold. The authors should alter the sentence to something along these terms: “although, to our knowledge, long-term recognition memory has (≧24 hours) findings have yet to be reported.”

Response 4) According to your comment, we have modified the sentence [Page 4, Line 83-84]

Point 5) The authors should consider including pictures of their testing environments and objects.

Response 5) Thank you for your suggestion. We have added Fig S1, which indicated pictures of each context and object used in our experiments.

Point 6) In the place and object versions, wouldn't an increase in retention performance be expected with the increasing sample phase durations? This is not the case and should be reconciled with the general hypothesis of the study. Also, relating this to the cited work by Dr. Mumby’s lab would be of benefit.

Mumby D, Gaskin S, Glenn M, Schramek T, Lehmann H. 2002. Hippocampal damage and exploratory preferences in rats: Memory for objects, places, and contexts. Learning & Memory 9: 49-57.

Rudy JW, Sutherland RJ. 1995. Configural association theory and the hippocampal formation: an appraisal and reconfiguration. Hippocampus 5: 375-389.

Saucier DM, Shultz SR, Keller AJ, Cook CM, Binsted G. 2008. Sex differences in object location memory and spatial navigation in Long-Evans rats. Anim Cogn 11: 129-137.

Response 6) Thank you for your comment. As you mentioned, our results did not show that the longer familiarization period simply improved recognition performance. We believe these results indicate that the appropriate familiarization periods for each recognition test are different, suggesting that an overlearning could affect recognition performance. To support this idea, we have added the results of statistical tests (Fig 2) and correlation analysis (Fig S2). And based on the Dr. Mumby’s lab work, we have added a paragraph [Page 14-15, Line 375-392] on the issue in Discussion.

Reviewer #2:

This series of experiments utilized the spontaneous objects recognition test used in rodents that can be used to assess recognition and more complex associative learning and memory processes. The specific question of interest for the investigators was whether altering the length of the sample phase would impact subsequent novelty discrimination in three different versions of the task including object, place, and object-in-place recognition tasks.

The results showed that increasing the familiarization period improved performance on the different variants of spontaneous recognition tasks, but this procedural manipulation was particularly important for the object-in-place task.

Overall, the experiments were well motivated and executed and the issue of optimal procedures for obtaining reliable results in learning and memory research is an important issue.

Response: We feel great thanks for your careful reading and interest in our article. According to your nice suggestions, we have made some corrections to our previous draft, the detailed corrections are listed below.

Point 1) What effect does the context pre-exposure have on learning these tasks?

Response 1) Thank you for your comment. We believe the context pre-exposure can facilitate the following learning. In a familiar environment, it is easy for rats to make an association between objects, locations, and contexts. We conducted the context pre-exposure before the object and place recognition tests. In these cases, we intended to eliminate the possibility that the preceding recognition tests could affect the following tests. We believe that the pre-exposure to the context without objects can contrast the subsequent recognition test with the preceding one. This contrast will allow rats to focus on a new pair of objects relative to the familiar information, such as backgrounds, in the sample phase.

Point 2) What effect does training the same animals on the three different versions? For example, if another investigator used the parameters used here but only one of the paradigms. Would the effects be the same?

Response 2) Thank you for your comment. As you have mentioned, we could not rule out the possibility that experience in the preceding test (OPC) could affect performance in the following tests (object or place recognition). In the Discussion, we have mentioned the limitation [Page 15, Line 400-408]. Nevertheless, we believe that our data of the OPC recognition test (Fig. 1D, 1F & Fig. 2) can be replicated in other laboratories since all the subjects experienced the OPC recognition test with no prior experience. In the present study, we focused on the effects of the familiarization length on performance in the more complex associative memory test (OPC). That is why we conducted the OPC test foremost. In addition, at least for the place recognition test, our data on the place recognition test are consistent with those of Ozawa et al. (2011), in which they conducted only place recognition test with three conditions of the familiarization period (5, 10, and 20 min). 

Ozawa T, Yamada K, Ichitani Y. Long-term object location memory in rats: effects of sample phase and delay length in spontaneous place recognition test. Neuroscience letters. 2011;497: 37–41. doi:10.1016/j.neulet.2011.04.022

Point 3) What about the task design found below? What impact does it have? “Rats assigned to one or another familiarization condition were subjected to the assigned condition in all tasks' sample phases.”

Response 3) Thank you for the comment. As you have mentioned, we should have counterbalanced the three familiarization conditions for each animal. However, to compare performance between OPC, place, and object recognition tests with a specific familiarization period directly, we adopted this task design. It is likely that the differences in time spent in the open-field arena in the proceeding recognition test cause a significant confounding effect on performance in the following one.

Point 4) line 130-where were the contexts placed (different rooms)? If in the same room, are they placed in different spots in the room. If not, is the path to and the entry to these rooms different (head direction).

Response 4) Thank you for the comment. Each open-field arena (white/ black) was located in a different place in an experimental room, but they were completely partitioned. Also, the white context was fully covered by a curtain, while the black one was not. To promote better understanding of these differences for readers, we have added photos of the apparatus [Fig S1].

Point 5) what is meant by place here? Egocentric space or more spatial/relational?

Response 5) The two open-field arenas used in our study have certain similarities in the size and direction of a spatial cue attached to the south side of the apparatus. Therefore, the element of place includes both egocentric and allocentric space meanings. The objects have the relational position between themselves and the absolute position by referring to the relationships between the rats’ location and the spatial cue. In the present study, the starting point of each trial was changed between the sample and test phases for each subject (this sentence have been added at [Page 5, Line 138-139]), which more demand to use a spatial cue (Langston & Wood, 2010). Langston & Wood (2010) developed a hippocampal-dependent version of the place recognition test that requires more recognition of the place element of allocentric information by changing the starting point of each trial.

Langston RF, Wood ER. Associative recognition and the hippocampus: Differential effects of hippocampal lesions on object‐place, object‐context and object‐place‐context memory. Hippocampus. 2010;20: 1139–1153. doi:10.1002/hipo.20714

Point 6) line 159-“Three-7” is a bit confusing.

Response 6) We have replaced it with a better one [Page 7, Line 162 and 173].

Point 7) statistics were appropriate and I liked the inclusion if inter-rater reliability.

Response 7) We appreciate your favorable comment. We have added an ANOVA analysis for Fig 2 and recalculated the inter-rater reliability in the revised revision since we have added Experiment 2.

Point 8) line 298-“leaning” should be changed to “learning”

Response 8) Thank you for your helpful comment. We have corrected the typographical error [Page 15, Line 402].

Point 9) line 299-“For example, Zeldin & Olton [23] demonstrated that prior spatial learning improved subsequent memory performance due to proactive interference.”

Why would previous spatial learning improve performance because of proactive interference? Unpack this for the reader.

Response 9) Thank you for your suggestion. The description of proactive interference was misleadingly incorrect. Thus, we have modified the description about a learning set [Page 15, Line 403-407] as below: “This effect is known as a learning set [23], in which acquiring a learning strategy in a preceding task improves performance in the following task. For example, Zeldin & Olton [24] demonstrated that prior spatial learning improved subsequent performance by developing a spatial learning set.”

Point 10) one difference between the tasks is the elements are the same on two versions but a completely novel element is introduced in the object recognition task. What impact does this have?

Response 10) Thank you for your suggestion. We totally agree with you that the novel element in the object recognition test is completely different from its in other recognition tests. Many studies (Langston & Wood, 2010; Chao et al., 2016; Wilson et al., 2013) have systematically investigated these elements of recognition memory within a subject. We directly compared the novel elements in parallel with these studies regardless of the effect as you mentioned.

Langston RF, Wood ER. Associative recognition and the hippocampus: Differential effects of hippocampal lesions on object‐place, object‐context and object‐place‐context memory. Hippocampus. 2010;20: 1139–1153. doi:10.1002/hipo.20714

Chao, O. Y., Huston, J. P., Li, J. S., Wang, A.-L., & Silva, M. A. de S. (2016). The medial prefrontal cortex—lateral entorhinal cortex circuit is essential for episodic‐like memory and associative object‐recognition. Hippocampus, 26(5), 633–645. doi: 10.1002/hipo.22547

Point 11) although mentioned in the discussion, the way that the task were trained sequentially in the same animals might be a significant confound in regards to the results. The concern is if the groups were only trained on one of these tasks with the experimental variable manipulations (sample time exposure) would you get the same effects?

Response 11) Thank you for your comment. If rats received three exploration conditions in a recognition test, manipulating the novelty would always be the same (e.g., one of the objects will be constantly replaced, or its position will constantly change). As the content of the novelty change becomes predictable for animals, their memory performance is expected to decline due to a discount in the novelty of the novel object.

---

## [Decision Letter · Decision Letter 1]

30 Jun 2021

Long-term associative memory in rats: effects of familiarization period in object-place-context recognition test

PONE-D-21-00442R1

Dear Dr. Yamada,

We’re pleased to inform you that your manuscript has been judged scientifically suitable for publication and will be formally accepted for publication once it meets all outstanding technical requirements.

Kind regards,

Robert Sutherland, Ph.D

Academic Editor

PLOS ONE

Additional Editor Comments (optional):

Reviewers' comments:

Reviewer's Responses to Questions

**Comments to the Author**

1. If the authors have adequately addressed your comments raised in a previous round of review and you feel that this manuscript is now acceptable for publication, you may indicate that here to bypass the “Comments to the Author” section, enter your conflict of interest statement in the “Confidential to Editor” section, and submit your "Accept" recommendation.

Reviewer #1: All comments have been addressed

Reviewer #2: All comments have been addressed

2. Is the manuscript technically sound, and do the data support the conclusions?

Reviewer #1: (No Response)

Reviewer #2: Yes

3. Has the statistical analysis been performed appropriately and rigorously? 

Reviewer #1: (No Response)

Reviewer #2: Yes

4. Have the authors made all data underlying the findings in their manuscript fully available?

Reviewer #1: (No Response)

Reviewer #2: Yes

5. Is the manuscript presented in an intelligible fashion and written in standard English?

Reviewer #1: (No Response)

Reviewer #2: Yes

6. Review Comments to the Author

Reviewer #1: (No Response)

Reviewer #2: (No Response)

7. PLOS authors have the option to publish the peer review history of their article (what does this mean?). If published, this will include your full peer review and any attached files.

Reviewer #1: No

Reviewer #2: **Yes: **Robert J. McDonald

---

## [Editor Report · Acceptance letter]

22 Jul 2021

PONE-D-21-00442R1 

Long-term associative memory in rats: effects of familiarization period in object-place-context recognition test 

Dear Dr. Yamada:

I'm pleased to inform you that your manuscript has been deemed suitable for publication in PLOS ONE. Congratulations! Your manuscript is now with our production department. 

Kind regards, 

on behalf of

Dr. Robert Sutherland 

Academic Editor

PLOS ONE